# Metabolite profile of *Nectandra oppositifolia* Nees & Mart. and assessment of antitrypanosomal activity of bioactive compounds through efficiency analyses

**Geanne A. Alves Conserva**[1☯], **Luis M. Quirós-Guerrero**[2,3☯], **Thais A. Costa-Silva**[1], **Laurence Marcourt**[2,3], **Erika G. Pinto**[4], **Andre G. Tempone**[5], **João Paulo S. Fernandes**[6], **Jean-Luc Wolfender**[2,3]*, **Emerson F. Queiroz**[2,3]*, **João Henrique G. Lago**[1]*

**1** Center of Natural Sciences and Humanities, Federal University of ABC, Santo Andre, Sao Paulo, Brazil, **2** School of Pharmaceutical Sciences, University of Geneva, Geneva, Switzerland, **3** Institute of Pharmaceutical Sciences of Western Switzerland, University of Geneva, Geneva, Switzerland, **4** Drug Discovery Unit, School of Life Sciences, University of Dundee, Dundee, United Kingdom, **5** Centre for Parasitology and Mycology, Adolfo Lutz Institute, São Paulo, Brazil, **6** Department of Pharmaceutical Sciences, Federal University of São Paulo, Diadema, São Paulo, Brazil

☯ These authors contributed equally to this work.
* jean-luc.wolfender@unige.ch (JLW); emerson.ferreira@unige.ch (EFQ); joao.lago@ufabc.edu.br (JHGL)

## Abstract

EtOH extracts from the leaves and twigs of *Nectandra oppositifolia* Nees & Mart. shown activity against amastigote forms of *Trypanosoma cruzi*. These extracts were subjected to successive liquid-liquid partitioning to afford bioactive $CH_2Cl_2$ fractions. UHPLC-TOF-HRMS/MS and molecular networking were used to obtain an overview of the phytochemical composition of these active fractions. Aiming to isolate the active compounds, both $CH_2Cl_2$ fractions were subjected to fractionation using medium pressure chromatography combined with semi-preparative HPLC-UV. Using this approach, twelve compounds (**1**–**12**) were isolated and identified by NMR and HRMS analysis. Several isolated compounds displayed activity against the amastigote forms of *T. cruzi*, especially ethyl protocatechuate (**7**) with $EC_{50}$ value of 18.1 μM, similar to positive control benznidazole (18.7 μM). Considering the potential of compound **7**, protocatechuic acid and its respective methyl (**7a**), *n*-propyl (**7b**), *n*-butyl (**7c**), *n*-pentyl (**7d**), and *n*-hexyl (**7e**) esters were tested. Regarding antitrypanosomal activity, protocatechuic acid and compound **7a** were inactive, while **7b-7e** exhibited $EC_{50}$ values from 20.4 to 11.7 μM, without cytotoxicity to mammalian cells. These results suggest that lipophilicity and molecular complexity play an important role in the activity while efficiency analysis indicates that the natural compound **7** is a promising prototype for further modifications to obtain compounds effective against the intracellular forms of *T. cruzi*.

**Data Availability Statement:** All relevant data are within the paper and its Supporting Information files.

**Funding:** 1. Fundação de Amparo a Pesquisa do Estado de São Paulo - FAPESP, projects 2018/07885-1, 2018/10279-6, 2015/23403-9, 2016/20633-6, and 2018/18975-1. 2. Conselho Nacional de Desenvolvimento Científico e Tecnológico (CNPq) for scientific research award to J.H.G.L and A.G.T. 3. Coordenação de Aperfeiçoamento de Pessoal de Nível Superior (CAPES) to fellow to T.A.C.S. L.M.Q.G 4. Ministerio de Ciencia, Tecnología y Telecomunicaciones, MICITT, from Costa Rica for the Scholarship provided (N° 214171-025). 5. Swiss National Science Foundation for the support in the acquisition of the NMR 600 MHz (SNF R'Equip grant 316030_164095).

**Competing interests:** The authors have declared that no competing interests exist.

# Introduction

The genus *Nectandra* is composed of more than 120 species with a wide diversity in the Brazilian Atlantic Forest [1]. The phytochemical profile of *Nectandra* species has shown a high variety of metabolites, including lignoids, alkaloids and terpenoids, among other compounds [2] with pharmacological potential [3–5]. As part of our continuous study aiming to discover antitrypanosomal natural products from plant biodiversity, our group has reported several promising compounds from these species. Recently, the antitrypanosomal neolignan licarin A was isolated from the *n*-hexane extract of *Nectandra oppositifolia* Nees & Mart. leaves and, considering its interesting activity against the amastigote forms of *Trypanosoma cruzi*, a series of semisynthetic derivatives were prepared for a structure-activity relationship (SAR) study [6]. Furthermore, three related butenolides were isolated from the *n*-hexane extract of *N. oppositifolia* twigs, which displayed anti-*Trypanosoma cruzi* activity [7] and immunomodulatory effect against *Leishmania infantum* [8]. These trypanosomatids are etiological agents of Chagas disease and leishmaniasis, respectively [9].

Chagas disease (CD) is an endemic disease in twenty-one countries of Latin America and, due to migration, occurrence has also increased in Europe and North America [10, 11]. The treatment of CD is still an important problem to be solved, since the approved drugs for the treatment (nifurtimox and benznidazole) exhibit several adverse side effects and high toxicity [12]. Furthermore, both drugs are effective only in the acute phase of the infection and inadequate in the chronic phase [12]. Therefore, the prospection of new compounds based on natural products is an important start point for development of new drugs for CD treatment.

As part of our efforts towards identification of antiprotozoal natural products [13–15], the EtOH extracts from the leaves and twigs of *N. oppositifolia* were investigated combining chemical dereplication of active fractions followed by isolation/identification of active compounds. Additionally, different derivatives were prepared and evaluated in order to investigate their antitrypanosomal activity with adequate drug-likeness by efficiency analysis.

# Materials and methods

## General experimental procedures

Optical rotations were measured in MeOH on a P-1030 polarimeter (Jasco, Japan). NMR spectra of compounds **1**–**12** were recorded on a Bruker Avance III HD 600 MHz NMR spectrometer equipped with a QCI 5 mm cryoprobe (Bruker BioSpin, Rheinstetten, Germany) using $CD_3OD$ (Aldrich) or $CDCl_3$ (Aldrich) as a solvent and internal standard. NMR spectra of compounds **7a**-**7e** were recorded on a Bruker Advance 300 MHz, using $CD_3OD$ (Aldrich) and $CDCl_3$ (Aldrich) as a solvent and TMS (Aldrich) as an internal standard. HPLC analyses were performed on 1260 infinity HPLC system equipped with a photodiode array detector (Agilent Technologies, Santa Clara, CA, USA) connected to an ELSD (evaporative light scattering detector) SEDEX Model FP (Sedere, France). UHPLC-QDa-ELSD analyses were performed on a Waters Acquity UPLC, equipped with PDA and QDA detectors, connected to an ELSD SEDEX Model FP. All used organic solvents were HPLC grade while $H_2O$ was purified using a Milli-Q system. Medium pressure liquid chromatography (MPLC-UV) was performed on a Sepacore® instrument (Buchi, Flawil, Switzerland) composed by pump module C-605, fraction collector model C-620 and UV spectrophotometer model C-640. Initial chromatographic procedures were performed using a flash $C_{18}$-HP column (200 x 30 mm I.D., 15 μm, Interchim, Montluçon, France). Semi-preparative HPLC-UV procedures were performed on a Shimadzu system equipped with an LC-20AP module pump, an SPD-20A UV/VIS, a 7725I Rheodyne® valve, and an FRC-10A fraction collector (Shimadzu, Kyoto, Japan), connected to

an ELSD SEDEX Model FP (Sedere, France), using loop and dry load injection system [16]. Semi-preparative separations were also performed using a Phenyl (250 mm x 21.2 mm I.D., 5μm, Phenomenex, CA, USA) and XBridge $C_{18}$ (250 mm x 10 mm, 5μm, Waters, MA, USA) column. Protocatechuic acid, MeOH, *n*-propanol, *n*-butanol, *n*-pentanol, *n*-hexaxol, were purchased from Aldrich with purity higher than 98% and used without further processing.

## Plant material

Fresh leaves and twigs from *N. oppositifolia* were collected at Artur Nogueira city, São Paulo State, Brazil (22˚30′57,65″S and 47˚10′50,11″W) in April/2016. The plant material was identified by Prof. MSc. Guilherme M. Antar and received a registration code at SISGEN A658372. A voucher specimen was compared with that under code SPF225339 deposited in the Herbarium of the Institute of Biosciences, University of São Paulo, SP, Brazil.

## Extraction

*N. oppositifolia* twigs (310 g) and leaves (332 g) were separately dried, macerated into powder and defatted with *n*-hexane (8 x 1.5 L each). After this procedure, the plant materials were extracted with EtOH 95% (10 x 2 L each) at room temperature. After solvent elimination under reduced pressure, 22.3 g and 94.1 g of crude EtOH extracts from the twigs and leaves, respectively, were obtained. Part of each EtOH extract (20.0 g) was individually ressuspended in MeOH:$H_2O$ 7:3 (v/v) and sequentially extracted using *n*-hexane and $CH_2Cl_2$. After solvent removal under reduced pressure, the respective *n*-hexane (2.65 g from twigs and 1.67 g from leaves) and $CH_2Cl_2$ (1.90 g from twigs and 1.69 g from leaves) phases were obtained.

## UHPLC-PDA-ESI-HRMS/MS

Chromatographic profiling of each $CH_2Cl_2$ phase (5 mg/mL) and isolated compounds (0.1 mg/mL) were performed on a Waters Acquity UHPLC system hyphenated to a Q-Exactive Focus mass spectrometer (Thermo Scientific, Bremen, Germany) using a heated electrospray ionization (HESI-II) source. The instrument was controlled using Thermo Scientific Xcalibur 3.1 software. LC separations were performed on a Waters BEH $C_{18}$ column (50 mm × 2.1 mm, 1.7 μm) using a linear gradient of 5−100% B over 7 min and an isocratic step at 100% B for 1 min. Mobile phases were (A): $H_2O$ with 0.1% formic acid, and (B): ACN with 0.1% formic acid at a flow rate of 600 μL min$^{-1}$ and an injection volume of 2 μL. ESI parameters were used as follows: source voltage, 3.5 kV; sheath gas flow rate ($N_2$), 55 units; auxiliary gas flow rate, 15 units; spare gas flow rate, 3.0; capillary temperature, 350˚C, and S-Lens RF Level, 45. The mass analyzer was calibrated using a mixture of caffeine, methionine–arginine–phenylalanine–alanine–acetate (MRFA), sodium dodecyl sulfate, sodium taurocholate, and Ultramark 1621 in ACN/MeOH/$H_2O$ solution containing 1% formic acid by direct injection. Data-dependent MS/MS events were performed on the three most intense ions detected in full scan MS (Top3 experiment). MS/MS isolation window width was 1 Da, and the stepped normalized collision energy (NCE) was set to 15, 30, and 45 units. In data-dependent MS/MS experiments, full scans were acquired at a resolution of 35,000 FWHM (at *m/z* 200) and MS/MS scans at 17,500 FWHM both with an automatically determined maximum injection time. After being acquired in a MS/MS scan, parent ions were placed in a dynamic exclusion list for 2.0 s.

## MS data treatment

All HRMS data was converted from RAW (Thermo) standard data format to.*mzXML* format using the MS Convert software, part of the ProteoWizard package [17]. The converted files

were treated using the MZmine software suite v. 2.53 [18]. For mass detection at $MS^1$ level a noise level of $1.0E^6$ for positive mode and $1.0E^4$ for negative mode were used. For $MS^2$ level 0.0E0 was set for both ionization modes. ADAP chromatogram builder was used and set to a minimum group size of scans of five, minimum group intensity threshold of $1.0E^6$ ($1.0E^4$ negative), the minimum highest intensity of $1.0E^6$ ($1.0E^4$ negative) and $m/z$ tolerance of 8.0 ppm. ADAP algorithm (wavelets) was used for chromatogram deconvolution. The intensity window S/N was used as S/N estimator with a signal to noise ratio set at 25, a minimum feature height at $1.0E^6$ ($1.0E^4$ negative), a coefficient area threshold at 100, peak duration range from 0.01 to 1.0 min, and the RT wavelet range from 0.01 to 0.08 min. Isotopes were detected using the isotope peaks grouper with an $m/z$ tolerance of 5.0 ppm, a RT tolerance of 0.02 min (absolute), the maximum charge set at 1, and the representative isotope used was the most intense. The resulting file was filtered using the peak-list rows filter option to remove features without an $MS^2$ associated spectrum.

## Mass spectral organization (Molecular networks) and taxonomically informed metabolite annotation

A molecular network was created from the.*mgf* file exported from MZmine for each ionization mode (positive and negative) using the online workflow (https://ccms-ucsd.github.io/GNPSDocumentation/) on the GNPS website (http://gnps.ucsd.edu) [19]. The precursor ion mass tolerance was set at 0.02 Da and a MS/MS fragment ion tolerance of 0.02 Da. A network was then created where edges were filtered to have a cosine score above 0.7 and more than four matched peaks. Spectra in the network were then searched against GNPS spectral libraries. The library spectra were filtered in the same manner as the input data. All matches kept between network spectra and library spectra were required to have a score above 0.7 and at least 4 matched peaks. The works are available in the following links, negative: https://gnps.ucsd.edu/ProteoSAFe/status.jsp?task=410aa8dbbd3146f480cd326d60a188cc; and positive: https://gnps.ucsd.edu/ProteoSAFe/status.jsp?task=4a3c00c5bd3344b0aacaf4152205cedc. Visualization of the results was performed using Cytoscape 3.8.0 software platform [20]. The GNPS output was used to annotate against the *in silico* ISDB-DNP [21] and then the script for taxonomically informed metabolite annotation [22] was used to re-rank and clean out the output based on the taxonomy.

## UHPLC-PDA-ELSD-MS analysis

Samples (5 mg/mL) were analyzed by UHPLC-PDA-ELSD-MS(Q). ESI conditions were as follows: capillary voltage 800 V, cone voltage 15 V, source temperature 120˚C, probe temperature 600˚C. Detection was performed in negative ion mode (NI) with $m/z$ range of 150–1000 Da. Chromatographic separations were performed on an Acquity UPLC BEH $C_{18}$ column (50 x 2.1 mm i.d., 1.7 μm; Waters, Milford, MA, USA) using mixtures composed by $H_2O$ (A) and ACN (B), both containing 0.1% formic acid, as the mobile phase, at 0.6 mL.min$^{-1}$ (40˚C). Analysis were conducted using the following gradient: from 5 to 100% of B from 0 to 7 min, 1 min at 100% B and a re-equilibration step of 2 min. ELSD was set at 45˚C, with a gain of nine. PDA data were acquired in the range of 190 to 500 nm, with a resolution of 1.2 nm. The sampling rate was set at 20 points/sec.

## Fractionation of bioactive $CH_2Cl_2$ fractions from twigs and leaves EtOH extracts of *N. oppositifolia*

Part of bioactive $CH_2Cl_2$ fraction (955 mg) from twigs EtOH extract of *N. oppositifolia* was subjected to flash chromatography using an Interchim $C_{18}$-HP column with $H_2O$ (0.1% formic

acid) (A) and MeOH (B) as the mobile phase. The elution was performed by using a linear gradient from 0 to 100% of B during 140 min. Initially, the $CH_2Cl_2$ fraction was mixed with 5 g of Zeoprep $C_{18}$ stationary phase (40–63 μm), and put into a dry-load cell (11.5 x 2.7 cm i.d.), which was connected between the pumps and column [23]. The flow rate was set to 20 mL min$^{-1}$, and the UV absorbance was monitored at 254 nm and 280 nm. This procedure afforded 57 fractions (50 mL each) which were pooled together into 25 groups (T1 –T25), after UHPLC-PDA-ELSD-MS analysis. Group T4 (6.0 mg) was purified by semi-prep. HPLC-UV in a phenyl column eluted with MeOH:$H_2O$ 85:15 (0.1% formic acid) to afford **1** (0.4 mg). Groups T6 (5.3 mg) and T18 (14.0 mg) were individually purified by semi-prep. HPLC-UV in a XBridge $C_{18}$ column eluted with MeOH:$H_2O$ 75:25 (0.1% formic acid) to afford **6** (0.8 mg) and **5** (0.1 mg), respectively. Groups T7 (28.0 mg), T8 (7.0 mg), and T10 (28.0 mg) were individually purified by semi-prep. HPLC-UV in a phenyl column eluted with MeOH:$H_2O$ 82:18 (0.1% formic acid) to afford, respectively, **9** (3.0 mg), **8** (0.7 mg) and **2** (0.2 mg). Group T9 (6.1 mg) was purified by semi-prep. HPLC-UV in a phenyl column eluted with MeOH:$H_2O$ 7:3 (0.1% formic acid) to afford **4** (1.3 mg). Group T14 (30.0 mg) was purified by semi-prep. HPLC-UV in an XBridge $C_{18}$ column eluted with MeOH:$H_2O$ 67:33 (0.1% formic acid) to afford **10** (0.2 mg) and **3** (0.2 mg).

Part of bioactive $CH_2Cl_2$ fraction (1.0 g) from the EtOH leaves extract of *N. oppositifolia* was subjected to MPLC chromatography using an Interchim $C_{18}$-HP column using the same conditions described for twigs $CH_2Cl_2$ fraction. This procedure yielded 53 fractions (50 mL each) which were pooled together into 10 groups (L1-L10), after UHPLC-QDa-ELSD analysis. Groups L3 (5.0 mg), L4 (7.5 mg), L6 (19.3 mg), and L8 (8.7 mg) were individually purified by semi-preparative HPLC-UV in an XBridge $C_{18}$ column eluted with MeOH:$H_2O$ 8:2 (0.1% formic acid) to afford, respectively, **7** (1.0 mg), **6** (0.2 mg), **11** (0.8 mg), **12** (1.4 mg), and **4** (1.4 mg).

## Synthesis of compounds 7a-7e

In a dry round-bottomed flask, protocatechuic acid (2 mmol, 0.308 g) and the respective alcohol (MeOH, *n*-propanol, *n*-butanol, *n*-pentanol, or *n*-hexanol—10 mL) were added. Immediately, 2 mL of $SOCl_2$ were added dropwise. The system was kept under reflux for 1 h and the volatiles were evaporated under reduced pressure. The residue was dissolved in 25 mL of EtOAc, washed with 2 x 15 mL aqueous $NaHCO_3$, and with 3 x 25 mL of $H_2O$. The organic phase was dried with anhydrous $Na_2SO_4$ and partially evaporated. The crude material was purified over silica gel column chromatography eluted with *n*-hexane:EtOAc 1:1 to afford **7a**-**7e** (for NMR and ESI-HRMS spectral data, see S1 File).

## Calculation of physical chemical properties and drug-likeness assessment

Calculation of physical chemical properties and prediction of pharmacokinetic potential for compounds **7** and **7a**-**7e** were done using the SwissADME online platform (http://www.swissadme.ch/). Table 4 summarizes the calculated properties, which includes logarithm of *n*-octanol/water (ClogP), fraction of sp$^3$ carbons (Fsp$^3$), hydrogen bond donor (HBD) and acceptor (HBA) group counts, molecular weight (MW), topological polar surface area (TPSA), rotatable bonds, and water solubility. The gastrointestinal (GI) absorption values were estimated using the correlation between ClogP and TPSA values as described in the SwissADME platform. The potential to inhibit CYP isoforms values were estimated using a prediction model described in the same platform [24].

The ClogP was considered the consensus value from SwissADME platform. The HBD and HBA counts are the number of OH's + NH's and O + N, respectively, in each molecule. The

Fsp$^3$ value is the number of sp$^3$ hybridized carbons divided by the total carbon count from the molecule. The water solubility criterion is described by the logS values in the SwissADME platform as follows: -2 > Soluble > -4 > moderately soluble > -6.

## Ligand metrics

The LE, LELP, LLE and FQ values were calculated for the bioactive compounds (**7**, **7b-7e**) by using Eqs 1–4 [25],

$$LE = 1.37(pEC_{50})/HA \tag{1}$$

$$LELP = ClogP/LE \tag{2}$$

$$LLE = pEC_{50}/ClogP \tag{3}$$

$$FQ = (pEC_{50}/HA)/0.0715 + (7.5328/HA) + (25.7079/HA^2) - (361.4722/HA^3) \tag{4}$$

where pEC$_{50}$ is the log($1/EC_{50}$) in molar unit and HA is the number of heavy (non-hydrogen) atoms in the respective compound.

## Experimental animals

The experimental animals (female BALB/c mice), used to obtain peritoneal macrophages, were obtained from the animal breeding facility at the *Instituto Adolfo Lutz*, São Paulo State, Brazil. BALB/c mice were kept in sterilized boxes with absorbent material in a controlled environment and received food and water *ad libitum*. Animal procedures were conducted with the approval of the Ethics Committee of *Instituto Adolfo Lutz* (project CEUA-IAL/Pasteur 05/2018) in accordance with the National Institutes of Health "Guide for the Care and Use of Laboratory Animals" (NIH Publications number 8023).

## Parasites and mammalian cell maintenance

*Trypanosoma cruzi* trypomastigotes (Y strain) were routinely maintained in *Rhesus* monkey kidney cells (LLC-MK2 –ATCC CCL 7), using RPMI-1640 medium supplemented with 2% FBS at 37˚C in a 5% CO$_2$-humidified incubator. The murine conjunctive cells (NCTC clone 929, ATCC) were maintained in RPMI-1640 medium supplemented with 10% FBS at 37˚C in a 5% CO$_2$ –humidified incubator. Peritoneal macrophages, used for the amastigote assays, were obtained by washing the peritoneal cavity of BALB/c mice with RPMI-1640 medium supplemented with 10% FBS and were kept at 37˚C in a 5% CO$_2$-humidified incubator.

## Determination of 50% effective concentration (EC$_{50}$) against *T. cruzi*

To evaluate the activity against intracellular amastigotes, the peritoneal macrophages obtained from BALB/c mice were infected with trypomastigotes forms of *T. cruzi*. The macrophages were plated on a 16-well chamber slide (NUNC plate, Merck; 1 x 10$^5$/well) and incubated for 24 h at 37˚C in a 5% CO$_2$-humidified incubator.

Sequentially, the trypomastigotes were washed in RPMI-1640 medium, counted, and used to infect the macrophages (parasite: macrophage, ratio = 10:1). After 2 h of incubation at 37˚C in a 5% CO$_2$-humidified incubator, the non-internalized parasites were removed by washing with RPMI-1640 medium. Compounds **1**–**12** and **7a**-**7e** were incubated by serial dilution (100–1.56 μM) with the infected macrophages (48 h at 37˚C, 5% CO$_2$) using benznidazole as the standard drug. At the end of the assay, slides were fixed with MeOH and stained with

Giemsa prior to counting under a light microscope (EVOS M5000, THERMO). The $EC_{50}$ values were calculated as previously described [26].

## Determination of cytotoxic concentration ($CC_{50}$) against mammalian cells

The cytotoxicity assays were performed using NCTC cells-clone 929. The cells were counted, seeded (6 x $10^4$ cells/well), and incubated with compounds **1**–**12** and **7a**- **7e** (200–1.56 μM—serial dilution) for 48 h at 37˚C in a 5% $CO_2$-humidified incubator. The optical density was determined using a FilterMax F5 (Molecular Devices) at 570 nm. The 50% cytotoxic concentration ($CC_{50}$) values were determined by MTT assay [27]. Selectivity index (SI) values were calculated using the ratio: $CC_{50}$ against NCTC cells/$EC_{50}$ against parasites form (amastigotes).

## Statistical analysis

$EC_{50}$ and $CC_{50}$ data represent the mean of two independent representative assays tested in duplicate and were calculated using sigmoid dose-response curves with Graph–Pad Prism 6.0 software (San Diego, CA, USA).

## Results and discussion

EtOH extracts from *N. oppositifolia* leaves and twigs exhibited activity against amastigote forms of *T. cruzi* (50% of parasite death at 300 μg/mL), indicating the presence of bioactive compounds. Sequentially, these extracts were individually subjected to successive partitions using *n*-hexane and $CH_2Cl_2$ to afforded inactive *n*-hexane and active $CH_2Cl_2$ (100% of parasite death at 300 μg/mL) fractions. Dereplication of both bioactive $CH_2Cl_2$ fractions was performed by UHPLC-PDA-HRMS/MS in positive (PI) and negative (NI) ionization modes. The HRMS data obtained was treated in MZmine [18] software and uploaded to the GNPS platform [19]. Two molecular networks were created (PI: 1902 nodes, NI: 1186 nodes) and subjected to an *in silico* ISDB-DNP dereplication process [21], followed by taxonomical informed metabolite annotation [22]. This process was restricted to compounds previously reported in Lauraceae and *Nectandra*, according to the Dictionary of Natural Products (DNP *v*28.2, http://dnp. chemnetbase. com/). Based on PI and of NI spectral data and spectrum ID, 19 compounds (**A–S**) were putatively identified in the studied extracts, as can be seen in Table 1 and Fig 1.

Three of the identified compounds are common organic acids/esters (**A**, **C** and **D**) and correspond to the most intense LC peaks detected on NI mode. As observed in Table 1, the other 16 compounds, corresponding to two alkaloids (**B** and **H**), twelve lignans (**E**, **F**, **I**-**O**, and **Q-S**), one phenylpropanoid (**G**), and one naphthalene derivative (**P**) have previously been reported in different species of *Nectandra* (*N. rigida*, *N. amazonum*, *N. puberula*, *N. turbacensis*, and *N. megapotamica*). Six of these compounds (**C**, **D**, **H**, **P-R**) exhibited an excellent match with the database from the GNPS platform, reinforcing their annotation confidence. Interestingly, licarin A and related alkenyl butenolides, previously isolated from *n*-hexane extracts of *N. oppositifolia* [6–8], were not detected in the studied EtOH extracts.

The presence of different structurally related unidentified metabolites in the bioactive studied phases justified an in-depth phytochemical investigation. Using medium pressure liquid chromatography (MPLC-UV) combined with semi-preparative HPLC-UV, it was possible to isolate compounds **1**–**12** (Fig 2) from the leaves and twigs of *N. oppositifolia*. These compounds were identified as ethyl (*R*)-pyroglutamate (**1**) [28], indole-3-aldehyde (**2**) [29], (+)-(*S*)-abscisic acid (**3**) [30], 2*Z*-(2,4-dihydroxy-2,6,6-trimethylcyclohexylidene)acetic acid (**4**) [31], azelaic acid (**5**) [32], vanillic acid (**6**) [33], ethyl protocatechuate (**7**) [34], scopoletin (**8**) [35], (-)-evofolin B (**9**) [36], moupinamide (**10**) [37], verrucosin (**11**) [38], and nectandrin B (**12**) [39], based on interpretations of optical rotation values, NMR, and ESI-HRMS spectra

**Table 1. Dereplicated compounds (ESI—positive and negative ionization modes) from CH$_2$Cl$_2$ phases of EtOH extracts obtained from *Nectandra oppositifolia* leaves and twigs.**

| ID | MF | Adduct type | Accurate mass | Error (ppm) | Rt (min) | Spectrum ID | Identified compound | Biological source |
|---|---|---|---|---|---|---|---|---|
| A | C$_7$H$_6$O$_3$ | [M-H]$^-$ | 137.0229 | -2.48 | 0.85 | FBS56-O[b] | 3-hydroxybenzoic acid | - |
| B | C$_{18}$H$_{19}$NO$_4$ | [M+H]$^+$ | 314.1384 | -0.81 | 1.02 | GNS51-A[b] | Isoboldine(S)-formN-De-Me | *N. rigida* |
| | C$_{18}$H$_{19}$NO$_4$ | [M-H]$^-$ | 312.1238 | 0.92 | 1.02 | GNS51-A[b] | Isoboldine(S)-formN-De-Me | *N. rigida* |
| C | C$_8$H$_8$O$_4$ | [M-H]$^-$ | 167.0336 | -1.41 | 1.38 | CCMSLIB00000578322[a] | (2,4-Dihydroxyphenyl)acetic acid | - |
| D | C$_9$H$_{10}$O$_4$ | [M-H]$^-$ | 181.0453 | -1.02 | 1.83 | MSV000081694[a] | 3,4,5-Trihydroxybenzaldehyde; 3,5-Di-Me ether | - |
| E | C$_{20}$H$_{24}$O$_5$ | [M+H]$^+$ | 345.1697 | 0.17 | 1.83 | HHF76-T[b] | 7,7'-Epoxy-3,3'-dimethoxy-4.4'-lignandiol (7R,7'R,8R,8'R)-form | *N. rigida* |
| F | C$_{20}$H$_{22}$O$_4$ | [M+H]$^+$ | 327.1593 | 0.19 | 1.83 | JDK65-I[b] | 4,7'-Epoxy-3,8'-lign-7-ene-3',4',5-triol(7E,7'R,8'R)-form3',5-Di-Me ether | *N. amazonum* |
| G | C$_{12}$H$_{18}$O$_5$ | [M-H]$^-$ | 241.1074 | -0.53 | 1.90 | QGC51-H[b] | 1-(3,4,5-Trihydroxyphenyl)-1,2-propanediol(1RS,2RS)-form3',4',5'-Tri-Me ether | *N. megapotamica* |
| H | C$_{18}$H$_{19}$NO$_4$ | [M-H]$^-$ | 312.1238 | -0.51 | 1.99 | CCMSLIB00005722482[a] | (E)-feruloyltyramine | *N. rigida* |
| I | C$_{20}$H$_{24}$O$_5$ | [M+H]$^+$ | 343.1521 | 0.61 | 2.26 | MCT77-G[b] | 3,3',4,4',7-Lignanpentol(8R,8'R)-form7-Ketone, 3,4-methylene, 3'-Me ether | *N. puberula* |
| J | C$_{21}$H$_{22}$O$_7$ | [M+H]$^+$ | 387.1438 | -1.81 | 2.41 | OQN88-W[b] | 7,9':7,9-Diepoxy-3,3',4,4',5-lignanpentol(7S,7'S,8R,8'R)-form3,4-Methylene, 3',5-di-Me ether | *N. turbacensis* |
| K | C$_{20}$H$_{20}$O$_6$ | [M+H]$^+$ | 357.1294 | 4.17 | 2.77 | GZM08-U[b] | 7,7'-Epoxy-3,3'-dimethoxy-4.4'-lignandiol (7R,7'R,8R,8'R)-form | *N. turbacensis* |
| L | C$_{20}$H$_{22}$O$_5$ | [M+H]$^+$ | 343.1517 | -4.28 | 3.07 | MCT77-G[b] | 3,3',4,4',7-Lignanpentol-(8R,8'R)-form7-Ketone, 3,4-methylene, 3'-Me ether | *N. puberula* |
| M | C$_{20}$H$_{22}$O$_5$ | [M+H]$^+$ | 343.1526 | -5.79 | 3.14 | MCT77-G[b] | 3,3',4,4',7-Lignanpentol(8R,8'R)-form7-Ketone, 3,4-methylene, 3'-Me ether | *N. puberula* |
| N | C$_{20}$H$_{22}$O$_5$ | [M+H]$^+$ | 343.152 | -7.57 | 3.20 | MCT77-G[b] | 3,3',4,4',7-Lignanpentol(8R,8'R)-form7-Ketone, 3,4-methylene, 3'-Me ether | *N. puberula* |
| O | C$_{21}$H$_{24}$O$_5$ | [M+H]$^+$ | 357.1674 | -7.94 | 3.24 | MCT82-E[b] | 3,3',4,4',7-Lignanpentol(8R,8'R)-form7-Ketone, 3,4-methylene, 3',4'-di-Me ether | *N. puberula* |
| P | C$_{20}$H$_{22}$O$_4$ | [M+H]$^+$ | 327.1588 | -7.19 | 3.34 | CCMSLIB00005721488[a] | 1-(4-hydroxy-3-methoxyphenyl)-6-methoxy-2,3-dimethyl-3,4-dihydro-1H-naphthalene-2,7-diol | *N. amazonum* |
| Q | C$_{21}$H$_{26}$O$_5$ | [M+H]$^+$ | 359.185 | -2.51 | 3.74 | CCMSLIB00000854329[a] | 4-[5-(3,4-Dimethoxyphenyl)-3,4-dimethyltetrahydro-2-furanyl]-2-methoxyphenol | *N. rigida* |
| R | C$_{22}$H$_{28}$O$_5$ | [M+H]$^+$ | 373.2002 | -3.82 | 4.14 | CCMSLIB00000081293[a] | Veraguensin | *N. puberula* |
| S | C$_{21}$H$_{24}$O$_5$ | [M-H]$^-$ | 355.1579 | 4.21 | 6.24 | MWS19-V[b] | Denudatin B(+)-form | *N. amazonum* |

[a] Spectral match with a spectrum from GNPS platform

[b] Spectral match with the *in silico* database, the code corresponding to that of CRC from the DNP.

(S1–S70 Figs in S1 File) associated with comparison of data previously reported in literature. This is the first occurrence of compounds **1–7, 9,** and **10** in the genus *Nectandra*.

Further exploration of the chemical classes obtained from ClassyFire [40] in the molecular networks, shown a good congruence in the clustering between the putatively identified and isolated compounds. For instance, compounds **11** and **12** clustered tightly with the putatively identified as furan neolignans **O-R**. Additionally, the consensus cluster class 'methoxyphenols' derived from the chemical ontology of ClassyFire [40] is in accordance with the proposed structures, as shown in Fig 3. To the best of our knowledge, both the putatively identified and isolated compounds are being reported for the first time at the species level.

Compounds **1–12** were evaluated against intracellular amastigote forms of *T. cruzi* and the determined EC$_{50}$ values are shown in Table 2. Compounds **4, 7, 9, 11** and **12** displayed moderate activities with EC$_{50}$ values of 28.2, 18.1, 27.8, 36.8, and 31.0 μM, respectively. Regarding the

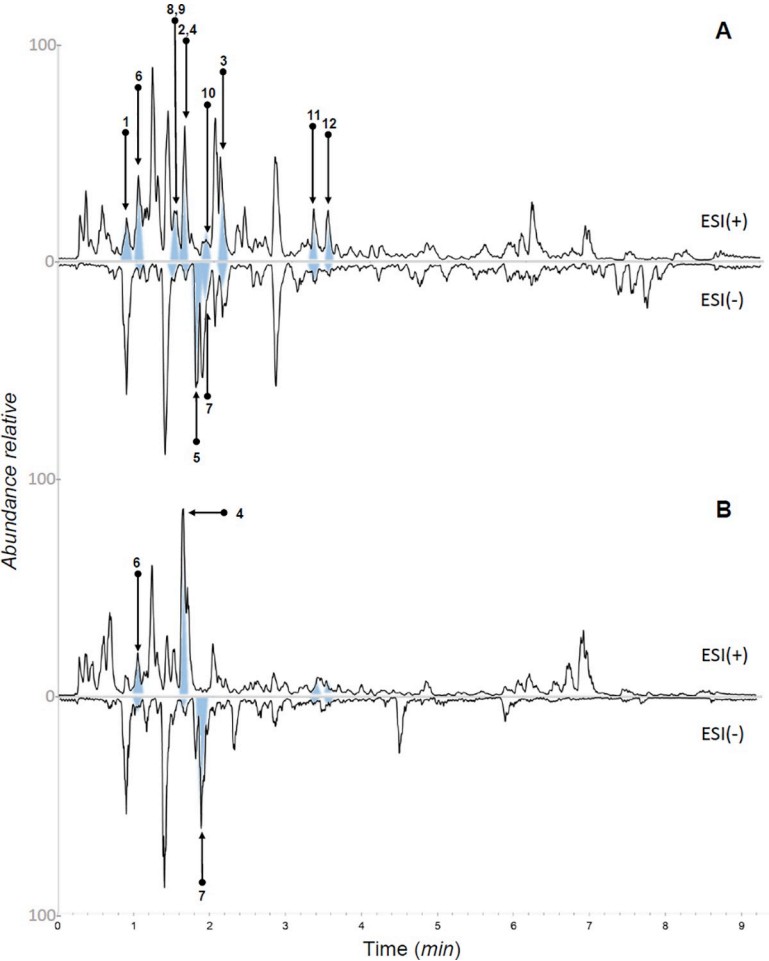

**Fig 1. UHPLC-PDA-ELSD-MS analysis of the bioactive CH₂Cl₂ fraction from EtOH extracts of *Nectandra oppositifolia* twigs (A) and (B) leaves in positive and negative ionization modes.** The corresponding isolated compounds (1–12) from each sample are indicated.

cytotoxicity against mammalian cells (NCTC), all tested compounds were non-toxic at the highest tested concentration (CC$_{50}$ > 200 μM), resulting in moderate selectivity index (SI) values (Table 2).

Among all isolated compounds, **7** (ethyl protocatechuate) exhibited a similar activity when compared to positive control benznidazole (EC$_{50}$ of 18.7 μM and SI > 10.7). Considering the activity of this compound in comparison to the phenolic acids **6** and **8**, it was suggested that the catechol moiety and the substitution in the free carboxylic acid to form an ester group play an important role in the activity against amastigotes. Previous reports demonstrated that *n*-octyl esters from vanillic and protocatechuic acids were active against *T. cruzi* amastigotes, with EC$_{50}$ values of 6.8 and 13.9 μM, respectively [41, 42], in accordance with the obtained results for compound **7**. Based on the DND*i* (Drug for Neglected Diseases initiative) criteria for the selection of new hits against amastigote forms of *T. cruzi* [43], compound **7** was selected for further investigation.

In order to assess the preliminary information regarding the relationship between structure and observed activity for this compound, a set of different esters (**7a-7e**) from protocatechuic acid were prepared and their *in vitro* potentials were evaluated. As observed in Table 3,

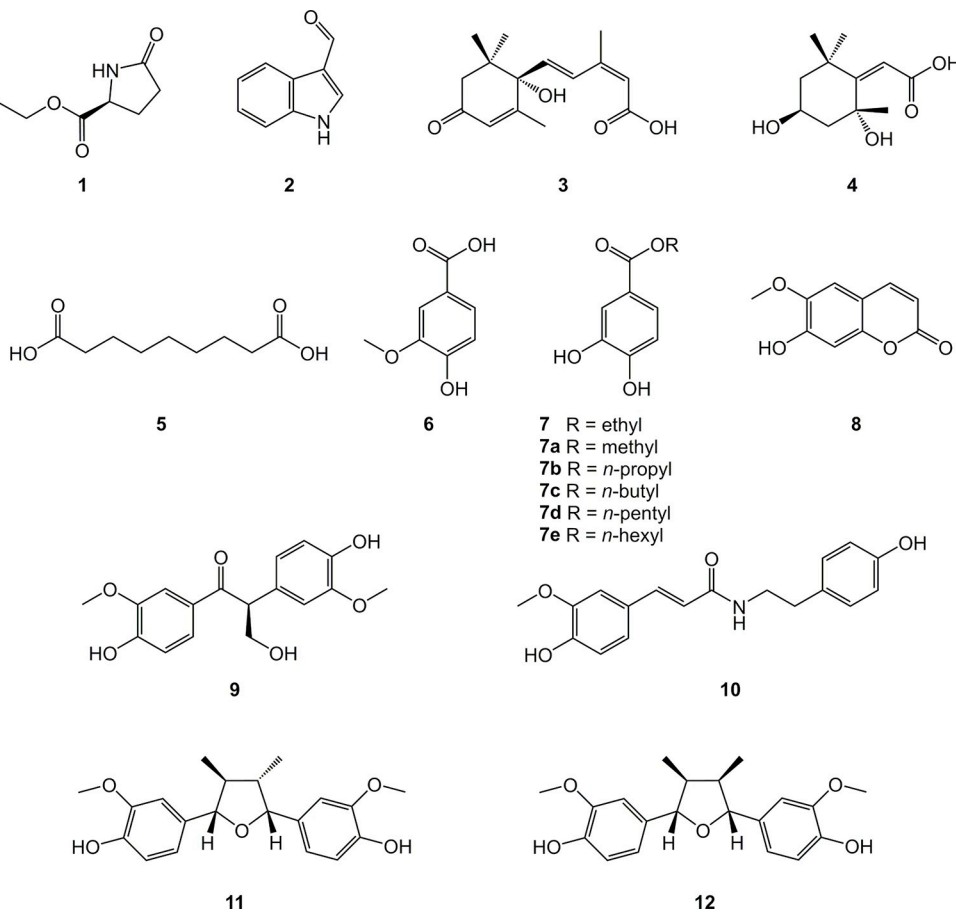

**Fig 2. Chemical structures of compounds 1–12 isolated from the bioactive CH₂Cl₂ fractions obtained from EtOH extracts of *Nectandra oppositifolia* leaves and twigs, and semi-synthetic derivatives 7a-7e.**

derivatives **7b**- **7e** displayed comparable or slightly higher activity than natural product **7** against the amastigote form of *T. cruzi*. On the other hand, the homologue **7a** (methyl ester) and protocatechuic acid were inactive. Interestingly, all tested compounds did not show appreciable cytotoxicity against mammalian cells, suggesting that the antiparasitic activity of the compounds is not correlated to unspecific cellular effect in the cells, leading to selective toxicity towards the parasite.

These results suggest that the presence of the ester group is essential to the antiparasitic activity of compound **7**, since the parent acid is completely inactive. Additionally, higher lipophilicity (as described by the ClogP value) increases the antiparasitic activity, suggesting that there is a minimal lipophilicity to assure the efficacy against *T. cruzi* amastigotes, in accordance with our previous report [42], where the *n*-octyl ester of protocatechuic acid showed activity against *T. cruzi* amastigotes. Interestingly, this increased lipophilicity did not affect the cytotoxicity profile of the compounds, reinforcing the hypothesis that the mechanisms behind the antiparasitic activity of these compounds are specific to the parasite cells.

The obtained results from SwissADME platform also suggest that the activity may be dependent of the unsaturation index (described as the ratio of $sp^3$ carbons to the total carbon atoms, $Fsp^3$), since this is the only parameter outside the "desirability radar" for the compound **7a** (Fig 4). As can be seen in Table 4, the addition of methylene units in the alkyl side chain increased the $Fsp^3$ ratio in parallel with the activity against *T. cruzi*. As reported in literature

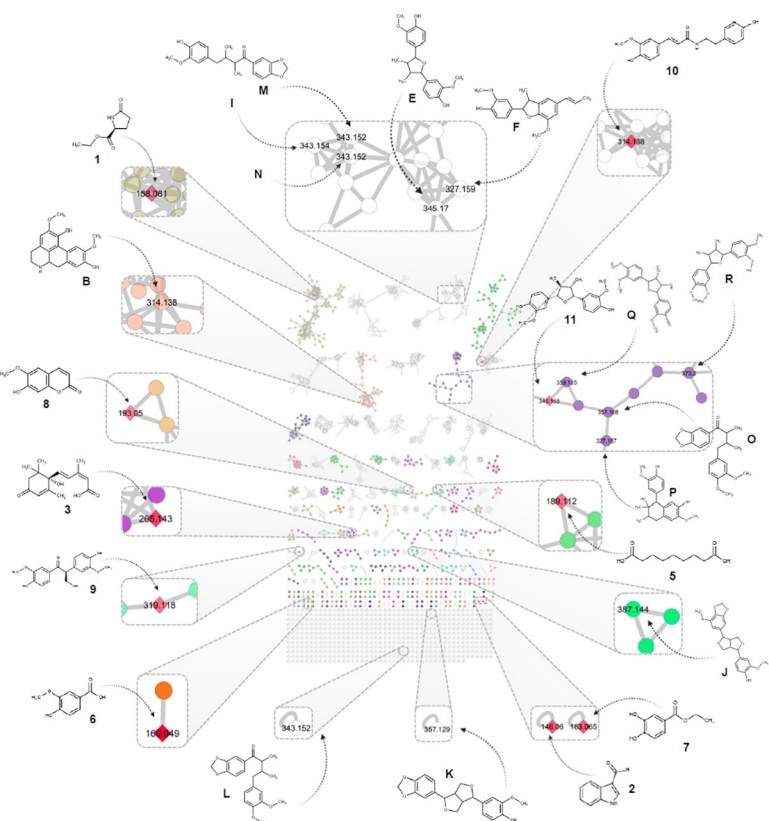

**Fig 3. Molecular network in positive ionization mode, generated from the UHPLC-HRMS/MS analysis of the CH₂Cl₂ fractions obtained from EtOH extracts of *Nectandra oppositifolia* leaves and twigs with a cosine similarity score cut-off of 0.6.** Chemical structures are shown in their respective nodes and clusters. Isolated compounds are represented by a red rhomboid (1–12). The putatively identified compounds (A-S) are also shown in the topology. Numbers inside the nodes correspond to the precursor mass for each feature. The edge's thickness is proportional to the similarity cosine between nodes. Several colors represent the different chemical consensus classes according to ClassyFire.

[44], the Fsp³ value describes the molecular complexity of a given compound and can be correlated to the drug-likeness considering that drugs present increased complexity when compared to non-drug compounds. In a previous report [7], the saturation degree (Fsp³) was observed to be related to the antitrypanosomal activity of butenolides isolated from *n*-hexane extract of *N. oppositifolia* twigs, especially against amastigote forms of *T. cruzi*. Isolinderanolide D, which contains a double bond in the side chain (C₁₆), displayed an EC₅₀ value of 25.3 μM while the saturated analogue isolinderanolide E showed an EC₅₀ of 10.1 μM.

As shown in Table 3, protocatechuic acid and its methyl ester (**7a**) displayed no activity at the highest tested concentration of 100 μM. However, ethyl (**7**), *n*-propyl (**7b**) and *n*-butyl (**7c**) esters exhibited EC₅₀ values of 18.1, 20.4 and 19.4 μM, respectively, demonstrating that the increment of molecular complexity and lipophilicity plays an important role in the activity. As expected, the more lipophilic and complex derivatives **7d** and **7e** showed increased activity, with EC₅₀ values of 12.3 and 11.7 μM, respectively (and consequently SI values > 16.3 and > 17.1).

The prediction using SwissADME platform indicates that all tested compounds may present high oral bioavailability, as predicted by the relationship between ClogP and TPSA values. Moreover, with exception of the more lipophilic compounds **7d** and **7e** (which showed potential inhibitory activity of CYP1A2 isoform) the prediction for these compounds did not

**Table 2. Anti-*Trypanosoma cruzi* activity of the compounds 1–12 isolated from the twigs and leaves of *Nectandra oppositifolia* and positive control benznidazole.**

| Compounds | $EC_{50} \pm SD$ / μM | $CC_{50}$ / μM | SI |
|---|---|---|---|
| **1** | NA | > 200 | - |
| **2** | NA | > 200 | - |
| **3** | NA | > 200 | - |
| **4** | 28.2 ± 2.9 | > 200 | > 7.1 |
| **5** | NA | > 200 | - |
| **6** | NA | > 200 | - |
| **7** | 18.1 ± 0.3 | > 200 | > 11.4 |
| **8** | NA | > 200 | - |
| **9** | 27.8 ± 0.2 | > 200 | > 7.2 |
| **10** | NA | > 200 | - |
| **11** | 36.8 ± 4.9 | > 200 | > 5.4 |
| **12** | 31.0 ± 0.3 | > 200 | > 6.4 |
| **Benznidazole** | 18.7 ± 4.1 | > 200 | > 10.7 |

$EC_{50}$–50% effective concentration, $CC_{50}$–50% cytotoxic concentration, SD–standard deviation, SI–selectivity index, calculated by the ratio $CC_{50}$ against NCTC cells/$EC_{50}$ against amastigote forms. NA–not active up to 100 μM. Benznidazole was used as positive control.

**Table 3. Anti-*Trypanosoma cruzi* activities of protocatechuic acid and esters 7, 7a-7e.**

| Compounds | $EC_{50} \pm SD$ / μM | $CC_{50}$ / μM | SI |
|---|---|---|---|
| protocatechuic acid | NA | > 200 | - |
| **7a** | NA | > 200 | - |
| **7** | 18.1 ± 0.3 | > 200 | > 11.4 |
| **7b** | 20.4 ± 4.2 | > 200 | > 9.8 |
| **7c** | 19.4 ± 6.3 | > 200 | > 10.3 |
| **7d** | 12.3 ± 3.8 | > 200 | > 16.3 |
| **7e** | 11.7 ± 2.7 | > 200 | > 17.1 |

$EC_{50}$–50% effective concentration, $CC_{50}$–50% cytotoxic concentration, SD–standard deviation, SI–selectivity index, calculated by the ratio $CC_{50}$ against NCTC cells/$EC_{50}$ against amastigote forms. NA–not active up to 100 μM.

**Table 4. Physical chemical, pharmacokinetic and drug-likeness prediction for the selected compounds.**

| Compounds | ClogP | $Fsp^3$ | HBD | HBA | MW | TPSA (Å$^2$) | Rotatable bonds | Solubility | GI absorption | CYP inhibition |
|---|---|---|---|---|---|---|---|---|---|---|
| Protocatechuic acid | 0.65 | 0.00 | 3 | 4 | 154 | 77.76 | 1 | Soluble | High | No |
| **7a** | 1.07 | 0.12 | 2 | 4 | 168 | 66.76 | 2 | Soluble | High | No |
| **7** | 1.40 | 0.22 | 2 | 4 | 182 | 66.76 | 3 | Soluble | High | No |
| **7b** | 1.77 | 0.30 | 2 | 4 | 196 | 66.76 | 4 | Soluble | High | No |
| **7c** | 2.09 | 0.36 | 2 | 4 | 210 | 66.76 | 5 | Soluble | High | No |
| **7d** | 2.46 | 0.42 | 2 | 4 | 224 | 66.76 | 6 | Moderate | High | CYP1A2 |
| **7e** | 2.67 | 0.46 | 2 | 4 | 238 | 66.76 | 7 | Moderate | High | CYP1A2 |

ClopP—logarithm of n-octanol/water; $Fsp^3$—unsaturation index, calculated by the ratio of sp$^3$ carbons to the total carbon atom; HBD—hydrogen bond donor (HBD) groups; HBA—hydrogen bond acceptor groups; MW–molecular weight; TPSA—topological polar surface area; GI—gastrointestinal absorption; CYP—cytochrome P450; CYP1A2—cytochrome P450 family 1 subfamily A member 2, involved in the metabolism of xenobiotics.

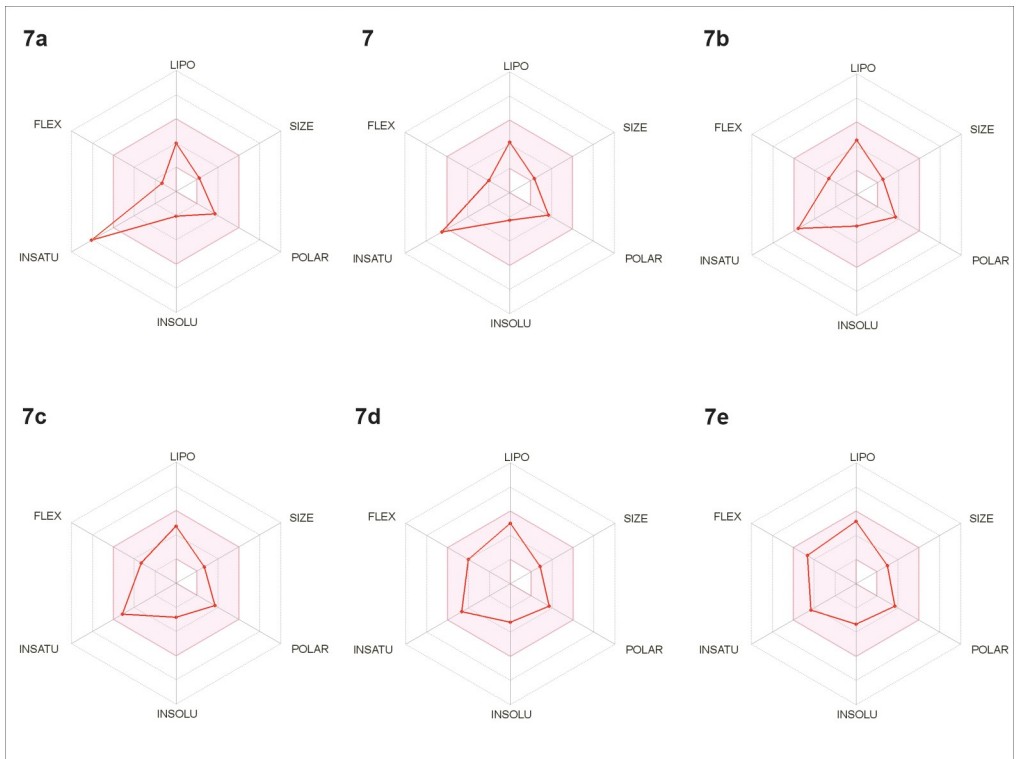

**Fig 4. *In silico* study of drug-likeness for derivatives 7, 7a – 7e, using the SwissADME platform.**

indicate important inhibitory activity for the CYP450 enzymes [24]. Regarding the drug-like-ness of the tested compounds, it must be stressed that none of these violated the Lipinski's Rule of Five, Veber's rules and other drug-likeness parameters, supporting the potential of such compounds as lead compounds with oral bioavailability [45]. Even the more lipophilic compound **7e** presented an ClogP value fairly below the limit for drug-likeness ($< 5$), as well as reduced molecular weight ($< 500$), indicating that there is still some space for further modifications in these prototypes.

Considering that during the optimization process in drug discovery the molecular complexity increases, the potency must be evaluated taking in account the molecular increments in the structure. Thus, it is expected that a more complex compound presents increased potency due to nonspecific (i.e. hydrophobic, van der Waals) binding interactions with the putative molecular target, which does not signify a real increment in potency [25]. In order to compare how the modifications in the protocatechuates impacted the biological activity, we carried out an analysis from calculated ligand efficiency metric values LE, LELP, LLE and FQ [25].

Ligand efficiency (LE) is a concept that correlates the potency with molecular weight and can be understood as a contribution of each heavy (non-hydrogen) atom of the molecule to the biological activity [25]. Literature considers that lead-like compounds may present LE $> 0.3$, while approved drugs usually present LE values $> 0.45$. As shown in Table 5, natural product **7** displayed the highest LE value in the set (0.50) and increase of the alkyl side chain caused a decrease of LE. This suggests that the alkyl chain is not directly related to a specific interaction with the putative molecular target in the parasite, but it may help to increase the penetration into the parasite cells. This is also corroborated by the fit quality (FQ) values, which is a size-independent LE modification. FQ values for all compounds were $> 0.54$, and the natural compound **7** was noteworthy due to the higher FQ value (0.57).

Table 5. Ligand metric analyses of the selected protocatechuates 7, 7b-7e.

| Compounds | LE | LELP | LLE | FQ |
|---|---|---|---|---|
| 7 | 0.50 | 2.80 | 3.34 | 0.57 |
| 7b | 0.46 | 3.86 | 2.92 | 0.55 |
| 7c | 0.43 | 4.86 | 2.62 | 0.54 |
| 7d | 0.42 | 5.85 | 2.45 | 0.55 |
| 7e | 0.40 | 6.72 | 2.26 | 0.55 |

LE—1.37($pEC_{50}$)/HA; LELP—ClogP/LE; LLE—$pEC_{50}$/ClogP; FQ—($pEC_{50}$/HA)/0.0715 + (7.5328/HA) + (25.7079/$HA^2$)–(361.4722/$HA^3$).

Literature data [46] suggests that the mean lipophilicity of approved drugs has been stable over the years. On the other hand, during the discovery process, the lipophilicity is substantially increased in hit-to-lead process, showing that lipophilicity directly affects the biological activity. However, excessive lipophilicity may favor unspecific binding interactions (such as hydrophobic interactions), what means that the increased activity observed for a given compound after molecular increments may not be correlated to specific interaction with the pharmacological target and can also promote off target interactions, leading to increased toxicity and inhibition of CYP450 enzymes. As an example, in previous work the *n*-octyl ester of protocatechuic acid [42] shown increased cytotoxicity ($CC_{50}$ 107.7 µM) in comparison to the esters (7, 7a-e) reported herein, which can be in part explained by the higher lipophilicity of this compound. Additionally, high lipophilicity impairs water solubility, which compromises the pharmacokinetic profile of drug candidates.

Considering this increment during the development process, lipophilicity-dependent metrics such as lipophilicity-corrected ligand efficiency (LELP) and lipophilic ligand efficiency (LLE) were derived from the LE concept. It is expected that adequate LELP values for a compound be in the -10 to +10 range, while adequate LLE values should be 5–7 [25]. Higher LLE values indicate that the lipophilicity of such a compound contributed significantly to the activity, while LLE values >7 indicate excessive lipophilicity. Once again, compound 7 must be highlighted since it presented the highest LLE value and the lowest LELP value, indicating that modifications during the optimization process, that increase the lipophilicity, may positively increase the antiparasitic activity. Moreover, the more lipophilic compounds 7d and 7e have already predicted inhibitory activity over CYP1A2 isoform (Table 4), corroborating the potential of compound 7 as a molecular prototype for further modifications.

## Conclusions

The present study reports the use of UHPLC-TOF-HRMS/MS, molecular network dereplication, and MPLC/HPLC purification procedures for detection/isolation of anti-*T. cruzi* metabolites from *N. oppositifolia*. This approach afforded twelve compounds (1–12), including the most active ethyl protocatechuate (7), with $EC_{50}$ value of 18.1 µM against amastigote forms of the parasite. Aiming to optimize its bioactivity, an *in silico* approach using SwissADME platform was performed and indicated that *n*-pentyl (7d) and *n*-hexyl (7e) derivatives demonstrated excellent adherence to all analyzed properties. As observed experimentally, these compounds showed higher antitrypanosomal activity against amastigotes in comparison to the positive control benznidazole. This data suggested that other structural modifications, maintaining the ethyl ester moiety, should be considered for future optimization of the studied compounds' activity.

## Supporting information

**S1 File. NMR and MS spectra for compounds 1–12 and 7a-7e are presented in S1-S70 Figs.**
(DOCX)

## Acknowledgments

This publication is part of the activities of the Research Network Natural Products against Neglected Diseases (ResNetNPND): http://www.uni-muenster.de/ResNetNPND/.

## Author Contributions

**Conceptualization:** Andre G. Tempone, Emerson F. Queiroz, João Henrique G. Lago.

**Data curation:** Geanne A. Alves Conserva, Thais A. Costa-Silva, João Paulo S. Fernandes, Emerson F. Queiroz, João Henrique G. Lago.

**Formal analysis:** Geanne A. Alves Conserva, Luis M. Quirós-Guerrero, Thais A. Costa-Silva, Laurence Marcourt, Andre G. Tempone, Jean-Luc Wolfender, Emerson F. Queiroz, João Henrique G. Lago.

**Funding acquisition:** Andre G. Tempone, Jean-Luc Wolfender, Emerson F. Queiroz.

**Investigation:** Geanne A. Alves Conserva, Luis M. Quirós-Guerrero, Thais A. Costa-Silva, Laurence Marcourt, Erika G. Pinto, Andre G. Tempone, João Paulo S. Fernandes, Jean-Luc Wolfender, Emerson F. Queiroz, João Henrique G. Lago.

**Methodology:** Geanne A. Alves Conserva, Luis M. Quirós-Guerrero, Thais A. Costa-Silva, Laurence Marcourt, Erika G. Pinto, João Paulo S. Fernandes, Jean-Luc Wolfender, Emerson F. Queiroz.

**Project administration:** Andre G. Tempone, Jean-Luc Wolfender, Emerson F. Queiroz, João Henrique G. Lago.

**Resources:** Luis M. Quirós-Guerrero, Laurence Marcourt, Andre G. Tempone, Jean-Luc Wolfender, Emerson F. Queiroz, João Henrique G. Lago.

**Software:** Luis M. Quirós-Guerrero, Laurence Marcourt, Erika G. Pinto, João Paulo S. Fernandes.

**Supervision:** Laurence Marcourt, Jean-Luc Wolfender, Emerson F. Queiroz, João Henrique G. Lago.

**Validation:** Geanne A. Alves Conserva.

**Visualization:** Geanne A. Alves Conserva, Luis M. Quirós-Guerrero, Laurence Marcourt, Andre G. Tempone.

**Writing – original draft:** Thais A. Costa-Silva, Laurence Marcourt, Andre G. Tempone, João Paulo S. Fernandes, Jean-Luc Wolfender, Emerson F. Queiroz, João Henrique G. Lago.

**Writing – review & editing:** Emerson F. Queiroz, João Henrique G. Lago.

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
