## [Decision Letter · Decision Letter 0]

30 Dec 2020

PONE-D-20-37405

Metabolite profile of Nectandra oppositifolia and assessment of the antitrypanosomal activity of bioactive compounds through efficiency analyses

PLOS ONE

Dear Dr. Lago,

Thank you for submitting your manuscript to PLOS ONE. After careful consideration, we feel that it has merit but does not fully meet PLOS ONE’s publication criteria as it currently stands. Therefore, we invite you to submit a revised version of the manuscript that addresses the points raised during the review process.

Several modifications of the text should be made according to the reviewers' suggestions.

Please add the species authority (Nees & Mart.) in the main title and the first mention in the Introduction section. Language should be polished by a native English speaker (e.g. L55: a genus cannot possess, it can contain...). Please do not use genus abbreviations in figure captions and table titles; whole Latin names shoud be provided instead. 

We look forward to receiving your revised manuscript.

Kind regards,

Branislav T. Šiler, Ph.D.

Academic Editor

PLOS ONE

Journal Requirements:

2. We noticed you have some minor occurrence of overlapping text with the following previous publications, which needs to be addressed:

- https://www.sciencedirect.com/science/article/abs/pii/S0045206819315536?via%3Dihub

- https://onlinelibrary.wiley.com/doi/abs/10.1002/elps.201900240

- https://www.mdpi.com/1660-3397/18/1/47/html

- https://www.frontiersin.org/articles/10.3389/fpls.2020.01287/full

In your revision ensure you cite all your sources (including your own works), and quote or rephrase any duplicated text.

Reviewers' comments:

Reviewer's Responses to Questions

**Comments to the Author**

1. Is the manuscript technically sound, and do the data support the conclusions?

Reviewer #1: Yes

Reviewer #2: Yes

2. Has the statistical analysis been performed appropriately and rigorously? 

Reviewer #1: Yes

Reviewer #2: Yes

3. Have the authors made all data underlying the findings in their manuscript fully available?

Reviewer #1: Yes

Reviewer #2: Yes

4. Is the manuscript presented in an intelligible fashion and written in standard English?

Reviewer #1: Yes

Reviewer #2: Yes

5. Review Comments to the Author

Reviewer #1: Introductive lines on potential of bioactive compounds and nutraceuticals should be adde d and related references added such as:

Durazzo, A.; Lucarini, M.; Santini, A. Nutraceuticals in Human Health. Foods 2020, 9, 370.

Lines 68-73 should be enlarged

Data in Table 1 should be better described in the text

Lines 367-373 shold be rewritten

Lines 484-490 should be better discussed.

Major details should be given in Tables 2, 3, 4, 5

Data in Figures 3 and 4 should be better described in the text

Limits, advantages, practical applications should be added in a paragraph Conclusion

The resolution of all Figures should be improved

Reviewer #2: The paper entitled " Metabolite profile of Nectandra oppositifolia and assessment of the antitrypanosomal activity of bioactive compounds through efficiency analyses” describe the dereprication of the bioactive fractions and the isolation of 12 known compounds. Compounds 7 was the most active against T. cruzi, and derivates were synthesized and tested.

Only few minor modifications are advised for acceptance:

1) Line 100, please correct the name of the city: Montluçon.

2) Line 360, please correct and of PI and NI data.

3) The quality of the figures 3 and 4 need to be improved.

4) Please add the position numbering on the structures to match the NMR data.

6. PLOS authors have the option to publish the peer review history of their article (what does this mean?). If published, this will include your full peer review and any attached files.

Reviewer #1: No

Reviewer #2: **Yes: **Ombeline Danton

---

## [Author Response · Author response to Decision Letter 0]

28 Jan 2021

EDITORIAL OFFICE COMMENTS

Thanks for your note – the revised version of this manuscript was adequately revised in order to meet the PLoS ONE style.

2. We noticed you have some minor occurrence of overlapping text with the following previous publications, which needs to be addressed:

• https://www.sciencedirect.com/science/article/abs/pii/S0045206819315536?via%3Dihub

• https://onlinelibrary.wiley.com/doi/abs/10.1002/elps.201900240

• https://www.mdpi.com/1660-3397/18/1/47/html

• https://www.frontiersin.org/articles/10.3389/fpls.2020.01287/full

Dear editor, thanks for this important note. As observed, these minor overlapping texts refer to Experimental Part, especially bioassays and efficient analysis. 

In your revision ensure you cite all your sources (including your own works), and quote or rephrase any duplicated text.

The manuscript was carefully revised to avoid duplicated text and the references were adequately cited. 

 

REVIEW COMMENTS TO THE AUTHOR

Reviewer #1

Introductive lines on potential of bioactive compounds and nutraceuticals should be added and related references added such as:

Durazzo, A.; Lucarini, M.; Santini, A. Nutraceuticals in Human Health. Foods 2020, 9, 370.

Thanks for your suggestion – this reference and related papers were added in the revised version of this manuscript.

Lines 68-73 should be enlarged

This sentence was modified accordingly.

Data in Table 1 should be better described in the text

More details concerning the analysis described in Table 1 were included in the revised version of this manuscript.

Lines 367-373 should be rewritten

These sentences were rewritten as suggested.

Lines 484-490 should be better discussed.

Thank you for this observation. This part of the manuscript was rephrased and the discussion was slightly modified. We hope that these modifications (highlighted in the revised version) had improved the discussion of this topic.

Major details should be given in Tables 2, 3, 4, 5

Additional information on Tables 2 – 5 were included as suggested.

Data in Figures 3 and 4 should be better described in the text

ClassyFire allows chemists to perform large-scale, rapid and automated chemical classification. Moreover, the accessible API allows easy access to more than 77 million “ClassyFire” classified compounds. The results can be used to help annotate known compounds as performed in the present work. Using this approach, as detailed indicated in figure 3, nineteen different and known compounds (A – S) were putatively identified in the studied extracts. Attending your suggestion, one reference was included in this part to allow to reads how these analyses were performed. 

In the case of figure 4, additional information was included in the revised manuscript, as suggested by the referee. 

Limits, advantages, practical applications should be added in a paragraph Conclusion

This sentence was modified accordingly as could be seen in the revised version of manuscript.

The resolution of all Figures should be improved

Attending your suggestion, the resolution of all figures was improved as could be seen in the revised version of the manuscript.

Reviewer #2

The paper entitled "Metabolite profile of Nectandra oppositifolia and assessment of the antitrypanosomal activity of bioactive compounds through efficiency analyses” describe the dereplication of the bioactive fractions and the isolation of 12 known compounds. Compound 7 was the most active against T. cruzi, and derivatives were synthesized and tested.

Only few minor modifications are advised for acceptance:

1) Line 100, please correct the name of the city: Montluçon.

Thanks for your careful reading and corrections in the manuscript. This point was modified accordingly in the revised manuscript.

2) Line 360, please correct and of PI and NI data.

Thanks again for your careful reading. This point corrected in the revised manuscript.

3) The quality of the figures 3 and 4 need to be improved.

The quality of all figures was improved attending your suggestion.

4) Please add the position numbering on the structures to match the NMR data.

Attending your suggestion, the structures of compounds 1 – 12 as well as the semisynthetic derivatives 7a – 7e, containing the position numbering, were included i

---

## [Editor Report · Decision Letter 1]

29 Jan 2021

PONE-D-20-37405R1

Metabolite profile of Nectandra oppositifolia and assessment of the antitrypanosomal activity of bioactive compounds through efficiency analyses

PLOS ONE

Dear Dr. Lago,

Thank you for submitting your manuscript to PLOS ONE. After careful consideration, we feel that it has merit but does not fully meet PLOS ONE’s publication criteria as it currently stands. Therefore, we invite you to submit a revised version of the manuscript that addresses the points raised during the review process.

The authors failed to spot and to respond to my comments provided in the previous report:

"Please add the species authority (Nees & Mart.) in the main title and the first mention in the Introduction section. Language should be polished by a native English speaker (e.g. L55: a genus cannot possess, it can contain...). Please do not use genus abbreviations in figure captions and table titles; whole Latin names shoud be provided instead."

Table 1 header: "source" instead of "sourct"

L565: "exhibited showed" - please remove one

Language improvement is still highly recommended.

We look forward to receiving your revised manuscript.

Kind regards,

Branislav T. Šiler, Ph.D.

Academic Editor

PLOS ONE

---

## [Author Response · Author response to Decision Letter 1]

29 Jan 2021

Branislav T. Šiler, Ph.D.

Academic Editor

PLOS ONE

Sao Paulo, January 29th, 2021

Dear Dr. Branislav T. Šiler,

There is included here the revised version of the manuscript Metabolite profile of Nectandra oppositifolia (Nees & Mart.) and assessment of the antitrypanosomal activity of bioactive compounds through efficiency analyses to be analyzed for publication in PLOS ONE (PONE-D-20-37405). 

The authors thank the suggestions and corrections made in the manucript, and we sincerely hope that new alterations we have made, and the extra material that we have added in response to the points will be satisfactory for publication in PLOS ONE. 

Sincerely yours

Prof. Dr. Joao Henrique G. Lago

---

## [Editor Report · Decision Letter 2]

2 Feb 2021

PONE-D-20-37405R2

Metabolite profile of Nectandra oppositifolia (Nees & Mart.) and assessment of the antitrypanosomal activity of bioactive compounds through efficiency analyses

PLOS ONE

Dear Dr. Lago,

Thank you for submitting your manuscript to PLOS ONE. After careful consideration, we feel that it has merit but does not fully meet PLOS ONE’s publication criteria as it currently stands. Therefore, we invite you to submit a revised version of the manuscript that addresses the points raised during the review process.

The authors have considerably improved the language usage throughout the text. However, minor flaws have still remained:

Species authority should not stand in brackets in the main title, L30 and L60 (please see e.g. http://www.plantsoftheworldonline.org/taxon/urn:lsid:ipni.org:names:281241-2).

Delete "(L.)" from the L65. Genus names are commonly abbreviated with its first letter; therefore it does not need clarification.

Please italicize the species Latin name in L215-216.

L261: "Table 6 summarizes" instead of "Table 6 summarize".

L548-550: The first sentence of the Conclusions section is vague, having poor syntax. Please rewrite to get sense.

We look forward to receiving your revised manuscript.

Kind regards,

Branislav T. Šiler, Ph.D.

Academic Editor

PLOS ONE

---

## [Author Response · Author response to Decision Letter 2]

4 Feb 2021

Dear Dr Anna Fodor

PLOS ONE

Recently, we received the following message from editorial office of PLOS ONE:

Your manuscript files have been checked in-house but before we can proceed we need you to address the following issues:

1) Please upload a copy of Supporting Information Figures S71-270 which you refer to in your text in line 387

Its important to mention that there is a mistake in this sentence - the correct is "Figures S1 - S70", as indicated in the Supporting information file. Please, the manuscript was adequately modified as suggested.

We sincerely hope that the new alterations we have made, and the extra material that we have added in response to the points raised by the referees, will be satisfactory for publication in PLOS ONE. 

Sincerely yours

Dr. Joao H G Lago

---

## [Editor Report · Decision Letter 3]

5 Feb 2021

Metabolite profile of Nectandra oppositifolia Nees & Mart. and assessment of the antitrypanosomal activity of bioactive compounds through efficiency analyses

PONE-D-20-37405R3

Dear Dr. Lago,

We’re pleased to inform you that your manuscript has been judged scientifically suitable for publication and will be formally accepted for publication once it meets all outstanding technical requirements.

Kind regards,

Branislav T. Šiler, Ph.D.

Academic Editor

PLOS ONE
---

## [Editor Report · Acceptance letter]

9 Feb 2021

PONE-D-20-37405R3 

Metabolite profile of *Nectandra oppositifolia* Nees & Mart. and assessment of antitrypanosomal activity of bioactive compounds through efficiency analyses 

Dear Dr. Lago:

I'm pleased to inform you that your manuscript has been deemed suitable for publication in PLOS ONE. Congratulations! Your manuscript is now with our production department. 

Kind regards, 

on behalf of

Dr. Branislav T. Šiler 

Academic Editor

PLOS ONE